# Evaluation of Plasma/Serum Adiponectin (an Anti-Inflammatory Factor) Levels in Adult Patients with Obstructive Sleep Apnea Syndrome: A Systematic Review and Meta-Analysis

**DOI:** 10.3390/life12050738

**Published:** 2022-05-16

**Authors:** Amir Najafi, Iman Mohammadi, Masoud Sadeghi, Annette Beatrix Brühl, Dena Sadeghi-Bahmani, Serge Brand

**Affiliations:** 1Oral and Maxillofacial Surgery Department, School of Dentistry, Isfahan University of Medical Sciences, Isfahan 8174673461, Iran; dr.amirnajafi@icloud.com (A.N.); iman_mohamadi59@yahoo.com (I.M.); 2Department of Biology, Science and Research Branch, Islamic Azad University, Tehran 1477893855, Iran; sadeghi_mbrc@yahoo.com; 3Center for Affective, Stress and Sleep Disorders, University of Basel, Psychiatric Clinics, 4001 Basel, Switzerland; annette.bruehl@upk.ch (A.B.B.); bahmanid@stanford.edu (D.S.-B.); 4Department of Psychology, Stanford University, Stanford, CA 94305, USA; 5Sleep Disorders Research Center, Kermanshah University of Medical Sciences, Kermanshah 6719851115, Iran; 6Substance Abuse Prevention Research Center, Kermanshah University of Medical Sciences, Kermanshah 6715847141, Iran; 7Department of Sport, Exercise and Health, Division of Sport Science and Psychosocial Health, University of Basel, 4052 Basel, Switzerland; 8School of Medicine, Tehran University of Medical Sciences, Tehran 1416753955, Iran

**Keywords:** obstructive sleep apnea syndrome, adiponectin, serum, plasma, meta-analysis

## Abstract

Background and objective: A variety of diseases, including obesity, type ‖ diabetes, and cardiovascular diseases are associated with obstructive sleep apnea syndrome (OSAS), and decreased adiponectin levels have been shown to be associated with an increased risk of these diseases. However, the association of blood levels of adiponectin in OSAS patients is a challenging and unknown issue with conflicting results. Therefore, we performed a systematic review and a meta-analysis to evaluate plasma/serum adiponectin levels in adult patients with OSAS. Materials and methods: A comprehensive search in four databases (PubMed/Medline, Web of Science, Scopus, and Cochrane Library) was performed in literature dated older than 12 March 2022, to retrieve the relevant articles. Effect sizes were calculated to show the standardized mean difference (SMD) along with a 95% confidence interval (CI) of plasma/serum of adiponectin between the OSAS patients and controls. The software RevMan 5.3, NCSS 21.0.2, CMA 2.0, trial sequential analysis (TSA) 0.9.5.10 beta, and GetData Graph Digitizer 2.26 were used for data synthesis in the meta-analysis. Results: A total of 28 articles including 36 studies were entered into the meta-analysis. The results showed that pooled SMD was −0.71 (95% CI: −0.92, 0.50; *p* < 0.00001; I^2^ = 79%) for plasma/serum levels of adiponectin in OSAS cases compared to the controls. The subgroup analyses showed that the geographical region and the Apnea-Hypopnea-Index (AHI) could be confounding factors in the pooled analysis of plasma/serum adiponectin levels. The sensitivity analysis showed the stability of the results. The radial and L’Abbé plots confirmed evidence of heterogeneity. Trial sequential analysis showed sufficient cases in the meta-analysis. Conclusions: With sufficient cases and stable results, the main finding of the meta-analysis identified significantly reduced plasma/serum levels of adiponectin in OSAS cases compared with the controls. This result suggests a potential role of adiponectin in the pathogenesis of OSAS.

## 1. Introduction

Obstructive sleep apnea syndrome (OSAS) is associated with recurrent episodes of upper airway obstruction during sleep [1,2] that can lead to increased negative intrathoracic pressure, sleep apnea, and intermittent hypoxia during sleep. Significant evidence suggests that OSAS is associated with metabolic disorders, inflammation, cardiovascular, neurocognitive disease, and cancer [1,3,4], involving hypertension, atrial fibrillation, diabetes, and pulmonary hypertension [1]. Therefore, OSAS affects the quality of life of individuals [5,6]. The OSAS is observed among more than 4% of the general population and among 35% to 45% of obese cases [7]. The diagnosis of OSAS is usually based on all-night polysomnography in the sleep laboratory, which collects a variety of physiological data during sleep [8]. The severity of OSAS is determined by the Apnea-Hypopnea Index (AHI = number of Apneas + Hypopneas/total sleep time) [9], with the following categories of AHI severity in adults: normal (AHI < 5), mild (AHI < 15), moderate (15 ≤ AHI < 30), and severe (AHI ≥ 30) [10].

The OSAS is a complex disease that may result from the interaction of several genetic and environmental factors [11]. The prevalence of OSAS is related to economic conditions, deep clinical effects on individual cognitive and overall function, and increased risk of adverse health complications [12]. The OSAS is strongly associated with cerebrovascular disorders, chronic neurological diseases, and inflammation, leading to a high risk of cognitive impairment in patients that continuous positive airway pressure (CPAP) therapy could improve [13]. Smoking [14], alcohol consumption [15], obesity [16,17], older age [18], and male gender [19] may act as other risk factors for OSAS. Several recent meta-analyses showed the association of genetic [20,21] and inflammation factors [22,23,24], hormones [25,26], and enzymes [27,28] with the risk of OSAS.

Adiponectin, also known as the 30 kDa fat supplement protein (Acrp30), is a protein hormone and adipokine that has been identified by various independent research groups [29]. This hormone is an endocrine agent that is synthesized and released from adipose tissue [30]. Basic scientific studies have shown that adiponectin has insulin-sensitive or anti-hyperglycemic [30], anti-atherogenic and anti-inflammatory properties [31,32]. Adiponectin may act as a protective and safe endocrine/paracrine/autocrine agent to prevent the development and/or progression of obesity-related fatal conditions [33]. 

Regarding the plasma/serum levels of adiponectin in individuals with OSAS compared to controls, the results have been mixed [34,35,36,37], with some studies reporting no differences of plasma/serum levels of adiponectin between individuals with and without OSAS, and some reporting lower plasma/serum levels of adiponectin in individuals with OSAS, compared to those without OSAS. Similarly, a meta-analysis [38] including data until January 2018 and based on 20 articles reported that the plasma/serum levels of adiponectin in individuals with OSAS were significantly lower than age-, sex-, and BMI-matched controls. We report a more recent meta-analysis to find a more conclusive answer to whether plasma/serum levels of adiponectin differed between individuals with and without OSAS, using a systematic review and meta-analysis based on 28 articles including trial sequential analysis (TSA) and radial and L’Abbé plots in individuals with OSAS, compared with age-, sex-, and BMI-matched controls, which had not been meta-analytically analyzed [38] so far.

## 2. Materials and Methods

### 2.1. Study Design

The present systematic review and meta-analysis was registered in PROSPERO (ID: CRD42022316318) and is in accordance with the Preferred Reporting Items for Systematic Reviews and Meta-Analyses (PRISMA) protocols [39]. The PECO (Population, Exposure, Comparator, and Outcome) question [40,41] was: are plasma/serum adiponectin levels different in individuals with OSAS compared to controls? (Humans with and without OSAS at any sex and age ≥ 18 years old: P; OSAS disease, E; OSAS patients compared to controls: C; and changes in the plasma/serum adiponectin levels: O).

### 2.2. Identification of Articles

One author (M.S.) performed a comprehensive search in four databases of PubMed/Medline, Web of Science, Scopus, and Cochrane Library for literature published by 12 March 2022 to retrieve the relevant articles. The search strategy was (“obstructive sleep apnea” or “sleep apnea” or “OSA” or “obstructive sleep apnea syndrome” or “OSAS” or “obstructive sleep apnea-hypopnea syndrome” or “OSAHS”) and (“adiponectin”) and (“plasma” or “blood” or “serum”). Moreover, the citations in the retrieved articles related to the subject were tested to ensure that no study was missed and then the titles and abstracts of the relevant articles were assessed by the same author (M.S.); subsequently, the full texts of the articles following the eligibility criteria were downloaded. Another author (A.N.) re-checked the process of the retrieved articles. A disagreement between two previous authors was resolved by a third author (S.B.).

### 2.3. Eligibility Criteria

Inclusion criteria were any study that: (1) included individuals with and without OSAS; (2) the participants had age ≥ 18 years old; (3) the individuals with OSAS were completely untreated; (4) reported plasma/serum adiponectin levels in individuals with and without OSAS; (5) the OSAS was diagnosed based on polysomnographic results; (6) the OSAS was defined as AHI ≥ 5 events/h; (7) the individuals with OSAS had no other systemic diseases; (8) the controls had no OSAS or any systemic disease. In contrast, exclusion criteria were: (1) meta-analyses, review articles, studies with incomplete data, studies without a control group or control groups with AHI ≥ 5 events/h; (2) studies including individuals younger than 18 years; (3) studies including individuals with OSAS with further somatic and psychiatric issues; (4) studies including individuals with sleep-disordered breathing; (5) conference papers, book chapters, and comment papers were also removed. 

### 2.4. Data Collection

Two authors (M.S. and I.M.) separately extracted the data of the articles included in the meta-analysis. Extracted data were the first author, the publication year, the country and ethnicity of participants, adiponectin source, assay approach, quality score, the sample size of individuals with OSAS, the mean levels of plasma/serum of adiponectin in samples with and without OSAS, the mean BMI and age, and the AHI of the samples with and without OSAS. 

### 2.5. Quality Evaluation

The quality evaluation was performed by one author (M.S.). The quality of the studies was evaluated using the Newcastle-Ottawa Scale (NOS) scale [42]. Nine was a maximum score for each study and a high-quality study had a score ≥7, a score = 6 was a moderate quality study, and a score ≤5 was a low-quality study.

### 2.6. Statistical Analysis 

The Review Manager software version 5.3 (RevMan 5.3; the Cochrane Collaboration, the Nordic Cochrane Centre, Copenhagen, Denmark) was used to compute the effect sizes mirroring the standardized mean difference (SMD) along with a 95% confidence interval (CI) of plasma/serum of adiponectin between individuals with OSAS and without OSAS. To estimate the pooled SMD significance, the Z-test was used, and a *p*-value (2-sided) less than 0.05 was considered significant. A P_heterogeneity_ < 0.1 or I^2^ > 50% showed significant heterogeneity and therefore a random-effects model [43], and if the heterogeneity was insignificant, a fixed-effect model [44] was utilized. 

The Galbraith (or Radial) plot shows the z-statistics (the result of division by standard error) on the vertical axis and the weight measurement on the horizontal axis [45], and the L’Abbé plot shows the event rate in the case group versus the event rate in the control group to help to explore the heterogeneity of effect estimates in a meta-analysis [46,47]. The plots were designed using the NCSS 2021 version 21.0.2 (NCSS, Kaysville, UT, USA) software.

A subgroup analysis (“Are the combined effect sizes in these subgroups significantly different from each other?”) was performed according to nine variables. 

A random-effects meta-regression analysis (describing a linear correlation between auxiliary variables in the study and effect size) was conducted based on seven variables. 

The degree of diffusion bias was determined using the funnel diagram and Egger’s regression test; Egger’s test is commonly used to evaluate the possible diffusion bias in a meta-analysis through funnel plot asymmetry. Egger’s test assesses a linear regression of the intervention effect estimates on their standard errors weighted by their inverse variance [48] and Begg’s test assesses if there is a significant relationship between the ranks of the effect estimates and the ranks of their variances [49]. The potential publication bias was tested in a Begg’s funnel plot by Begg’s test and the degree of asymmetry was examined by Egger’s test. The *p*-values of Egger’s and Begg’s tests were extracted and a *p*-value (2-sided) less than 0.10 recommended the existence of the publication bias. The trim-and-fill method was used to estimate potentially missing studies due to publication bias in the funnel plot and adjusting the overall effect estimate.

To evaluate the stability/consistency of pooled SMDs, both “one-study-removed” and “cumulative” analyses were used.

Publication bias and sensitivity analyses were carried out applying the Comprehensive Meta-Analysis software version 2.0 (CMA 2.0; Biostat Inc., Englewood, NJ, USA).

To address false-positive or negative conclusions from meta-analyses [50], TSA was conducted using TSA software (version 0.9.5.10 beta) (Copenhagen Trial Unit, Centre for Clinical Intervention Research, Rigshospitalet, Copenhagen, Denmark) [51]. A futility threshold could be examined by TSA to achieve a result of no impact before achieving the information size. The required information size (RIS) with an alpha risk of 5%, a beta risk of 20%, and a two-sided boundary type was computed. We estimated D^2^ as 91% for the plasma/serum levels of adiponectin, and the mean difference (MD) and variance were based on empirical assumptions that were autogenerated by the software. If the Z-curve reached the RIS line or monitored the boundary line or futility area, enough cases were included in the studies and the conclusions were reliable. Otherwise, the amount of information was not large enough and there was a requirement for further evidence. 

In some studies, the data were extracted from a graph using GetData Graph Digitizer software version 2.26.0.20 (GetData Pty Ltd., Kogarah, Australia). All authors examined the final statistical analyses, and any concerns were resolved by a discussion.

## 3. Results

### 3.1. Study Selection

The search in four databases and by hand in other sources identified 588 records (Figure 1). After removing duplicates and irrelevant records, 96 full-text articles were assessed based on the criteria above: 68 articles were excluded for various reasons (1 was a meta-analysis, 4 were reviews, 4 reported polymorphisms of adiponectin, 3 reported patients with sleep-disordered breathing (SDB), 13 reported CPAP therapy on OSAS patients, 15 lacked a control group or the control group had AHI > 5 event/h, 1 reported OSAS cases with Launois–Bensaude Syndrome, 1 reported OSAS cases with non-alcoholic fatty liver disease, 1 reported OSAS cases with erectile dysfunction, 1 reported OSAS cases with cerebral infarction, 7 reported child individuals, 11 did not match one or more factors (age, sex, and BMI) between cases and controls, 1 did not report any data for adiponectin level in the control group, 4 had no data for calculating the mean, and 1 reported mean without a standard deviation). Finally, 28 articles including 36 studies were entered into the meta-analysis.

### 3.2. Characteristics

Table 1 and Table 2 show the characteristics of included articles and studies in the meta-analysis, respectively. The articles were published from 2004 to 2020. Ten articles [37,52,53,54,55,56,57,58,59,60] were reported from China, four [36,61,62,63] from the USA, four [64,65,66,67] from Turkey, three [68,69,70] from Brazil, one [71] from India, one [72] from Japan, one [73] from Korea, one [74] from Spain, one [35] from Kuwait, one [75] from Italy, and one [34] from Egypt. Thirteen articles [37,52,53,54,55,56,57,58,59,60,71,72,73] identified the individuals as Asians, eight [34,35,64,65,66,67,74,75] as Caucasians, and seven [36,61,62,63,68,69,70] as mixed ethnicity. Eighteen articles [34,36,37,52,53,55,56,57,58,59,61,62,64,65,66,67,72,75] reported serum and ten [35,54,60,63,68,69,70,71,73,74] reported salivary levels of adiponectin. With regards to the assay approach, nineteen articles [34,35,37,55,56,57,58,59,60,61,62,63,66,67,69,70,71,72,75] measured adiponectin levels with enzyme linked immunosorbent assay (ELISA) and nine [36,52,53,54,64,65,68,73,74] measured adiponectin levels with radioimmunoassay (RIA). The quality score based on the NOS for each study is shown in Table 1.

The characteristics of the subjects in the included studies in the meta-analysis were number of cases and controls and mean levels of adiponectin, age, BMI, and AHI for cases and controls.

### 3.3. Pooled Analysis

A pooled analysis showed that plasma/serum adiponectin levels in the OSAS cases were lower than the controls (Figure 2). The pooled SMD was −0.71 (95% CI: −0.92, 0.50; *p* < 0.00001; I^2^ = 79%). 

### 3.4. Subgroup Analysis

A subgroup analysis was performed based on nine variables (Table 3). Among variables, the country of the individuals (region) and AHI in cases could be confounding factors in the pooled analysis of plasma/serum adiponectin levels in OSAS cases versus controls.

### 3.5. Sensitivity Analysis

A sensitivity analysis based on “one study removed (Figure 3) analysis” and “cumulative analysis” (Figure 4) showed the stability of the pooled analysis.

### 3.6. Trial Sequential Analysis

The results showed that the cumulative Z-curve (blue line) has successfully crossed the conventional boundary (Z-statistic above 1.96), the trial sequential monitoring boundary (concave red line), and RIS line (vertical red line) for plasma/serum levels of adiponectin (Figure 5). Therefore, the TSA showed sufficient evidence supporting the finding of lower plasma/serum levels of adiponectin in the OSAS patients compared to controls (the conclusion is reliable and has sufficient cases) and therefore further relevant studies are unnecessary. 

### 3.7. Meta-Regression

A random-effects meta-regression of plasma/serum levels of adiponectin for individuals with OSAS versus controls is shown in Table 4. The results showed that publication year, sample size, age, AHI, and BMI were moderator factors in the pooled analysis.

### 3.8. Heterogeneity 

Figure 6 illustrates the Galbraith (Radial) plot of plasma/serum adiponectin levels for individuals with OSAS versus controls. The plot detected 16 potential outliers that were removed and after that, the pooled SMD became −0.34 with lack of heterogeneity (95% CI: −0.47, −0.22; *p* < 0.00001; I^2^ = 4%). Therefore, these outliers can be one of the sources of heterogeneity in the initial analysis.

Figure 7 shows the L’Abbé plot of the plasma/serum adiponectin levels for individuals with OSAS versus the controls. The plot displays evidence of heterogeneity (Q statistic, 229.90; *p* < 0.001). Therefore, a significant relationship between the mean levels of adiponectin in individuals with and without OSAS appears to be unlikely.

### 3.9. Publication Bias

Figure 8 shows Begg’s funnel plot. The Egger’s (*p* = 0.052) and Begg’s (*p* = 0.022) tests would reveal a publication bias for the plasma/serum levels of adiponectin in individuals with vs. without OSAS (*p*-values of both tests < 0.10).

Table 5 illustrates the results of the trim-and-fill method on bias. For the plasma/serum levels of adiponectin and 0 imputed studies, under the fixed-effects model, the point estimate and pseudo 95% CI for the combined studies was −0.647 (−0.740, −0.553); using the trim-fill method, the imputed point estimate was −0.647 (−0.740, −0.553). In addition, under the random-effects model, the point estimate and 95% CI for the combined studies was −0.726 (−0.939, −0.513); using the trim-fill method, the imputed point estimate was −0.726 (−0.939, −0.513). Therefore, the overall effect sizes on the plasma/serum levels of adiponectin reported in the forest plot appeared valid, with a trivial publication bias effect based on fixed-effects and random-effects models, because the observed estimates were similar to the adjusted estimates.

## 4. Discussion

The present meta-analysis with sufficient cases and stable results suggests that the plasma/serum adiponectin levels in individuals with OSAS were lower than those in age-, sex-, and BMI-matched individuals without OSAS. Though, due to outliers, a high heterogeneity was also observed. Further, the subgroup analysis showed that geographical area and AHI in individuals without OSAS could be a confounding factor in the pooled results. Several studies [53,65,66,75] have shown that OSAS was a potential driver of decreased levels of adiponectin independently from age, gender and BMI, and thus similar to what has been observed in the present meta-analysis.

Adiponectin is a messenger that binds adipose tissue to other organs and is believed to play a key role in insulin resistance, especially in type 2 diabetes. In one study [35], low adiponectin was associated with OSAS severity and the association between insulin resistance and OSAS could be explained by differences in BMI, age, and sex. In addition, the fasting blood glucose (FBG) and the homeostasis model assessment of insulin resistance (HOMA-IR) also increase in parallel to the OSAS severity [35]. Another study [76] showed that the OSAS severity was related to the HOMA-IR. The association between plasma/serum levels of adiponectin and OSAS severity was confirmed in several studies [67,77]; likewise, the present meta-analysis showed that AHI can be a risk factor for OSAS.

The effect of obesity on adiponectin levels in OSAS patients in the studies led to different results. Masserini et al. [78] reported that the serum adiponectin level was significantly decreased in obese patients with OSAS in comparison with healthy normal-weight controls. Zidan et al. [77] showed a significant negative correlation between serum adiponectin levels and BMI and waist circumference in patients with OSAS. Nakagawa et al. [72] found that adiponectin levels in OSAS patients were only significantly related to the waist to hip ratio. In the present meta-analysis, BMI had no effect on the association between adiponectin levels and the risk of suffering from OSAS.

Decreased adiponectin concentrations may both exert a detrimental impact on cardiovascular homeostasis and, may be an independent risk factor for cardiovascular issues [79,80,81,82,83]. In contrast to several studies included in the present meta-analysis, two studies [36,74] showed an elevated plasma level of adiponectin in individuals with OSAS, compared to individuals without OSAS. These results suggest that although OSAS is known to be a risk factor for cardiovascular issues [84,85], this disease may trigger some “compensatory” processes that can be thought of as protecting the cardiovascular system.

López-Jaramillo et al. [86] reported regional differences in plasma adiponectin levels in cases with metabolic syndrome. A meta-analysis [87] showed that geographic region could affect the levels of circulating adiponectin in cases with prediabetes, as there was a significantly decreased adiponectin level in cases with prediabetes compared to controls in Asia and Europe, but not for the USA. The present meta-analysis showed different results in Chinese cases compared to American, Turkish, and Brazilian cases. There was an association between adiponectin levels and the risk of OSAS just for the Chinese cases. 

The meta-regression analysis in a meta-analysis conducted by Lu et al. [38] on individuals with and without OSAS showed that the adiponectin levels were significantly correlated with race, whereas publication year, adiponectin source, assay approach, mean age, BMI, and AHI were not. In our meta-analysis, the meta-regression showed publication year, sample size, AHI, BMI, and mean age had no relationship with the plasma/serum adiponectin levels. Further, the subgroup analysis in our meta-analysis showed that ethnicity, country (region), adiponectin source, assay approach, sample size, publication year, AHI, BMI, and mean age had no effect on the pooled results of plasma/serum adiponectin levels in individuals with and without OSAS; similarly, the subgroup analysis of Lu et al. [38] had identified the same pattern.

Despite the robust pattern of results, the following limitations should be considered: (1) Although we used a comprehensive search strategy, we could not claim that all available studies were extracted, as there may have been studies in different languages and in local databases and sources that we did not have access to. (2) There were some conference abstracts that might have been included, but they did not report available data. (3) A bias and heterogeneity across or among the studies can create confounding factors in the results. (4) There were different designs (cross-sectional, case-control, cohort, etc.) among the studies included in the present meta-analysis. (5) Most studies were published prior to 2018, and therefore they may have been included in previous meta-analysis. Therefore, earlier studies will effectively have a greater weighing to our meta-analysis.

In contrast, the following strengths are highlighted: there were a sufficient number of studies and samples of individuals with and without OSAS; the results were stable; there were matching age, gender, and BMI between samples of individuals with and without OSAS.

## 5. Conclusions

The main findings of the meta-analysis with sufficient cases and stable results showed significantly decreased plasma/serum levels of adiponectin in individuals with OSAS, compared to individuals without OSAS. This result suggests a potential role of adiponectin in the pathogenesis and maintenance of OSAS; such decreased adiponectin levels might also explain the high incidence of cardiovascular issues and the metabolic syndrome among individuals with OSAS. Future studies, of course, can be performed with an emphasis on geographical area, race, and severity of OSAS or AHI to further determine the effect of these interfering factors. Further such studies might better explain the pathophysiological process affecting the plasma/serum adiponectin levels in OSAS cases. Last, interventional studies with CPAP devices might give the possibility to investigate, if plasma/serum levels of adiponectin levels might increase, once the CPAP device was well applied and established. 

## Figures and Tables

**Figure 1 life-12-00738-f001:**
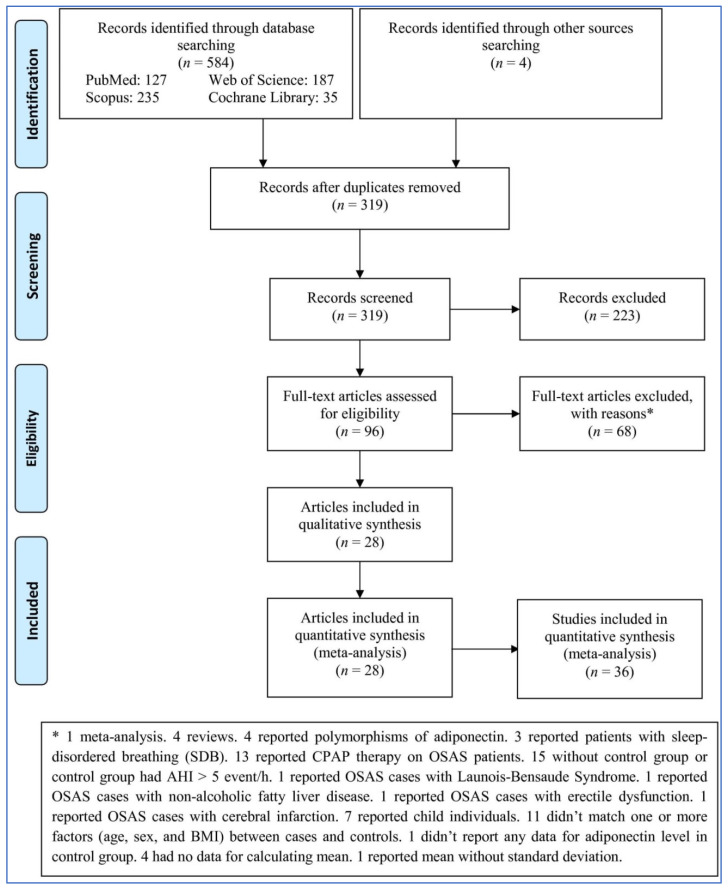
Flowchart of the meta-analysis.

**Figure 2 life-12-00738-f002:**
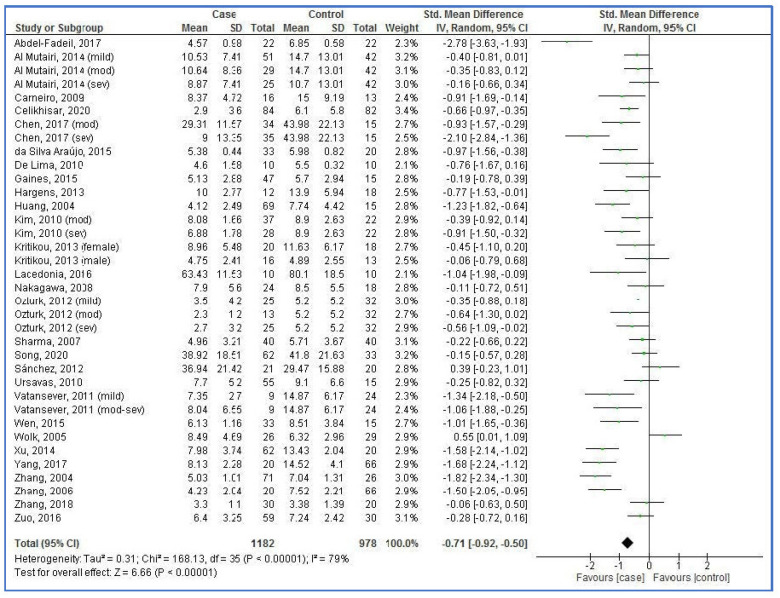
Forest plot of studies on plasma/serum adiponectin levels for obstructive sleep apnea syndrome cases versus controls [34,35,36,37,52,53,54,55,56,57,58,59,60,61,62,63,64,65,66,67,68,69,70,71,72,73,74,75]. SD, standard deviation; CI, confidence interval.

**Figure 3 life-12-00738-f003:**
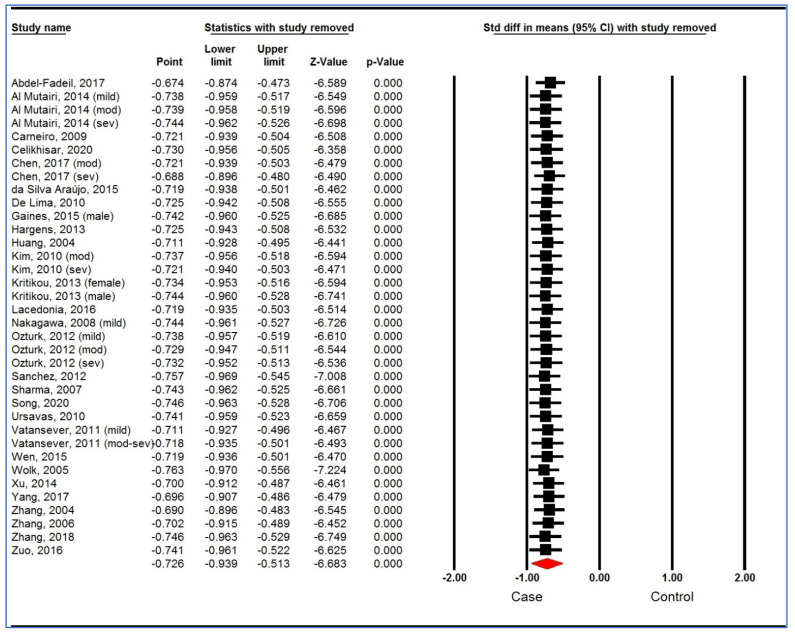
Sensitivity analysis (one study removed analysis) of studies on plasma/serum adiponectin levels for obstructive sleep apnea syndrome cases versus controls [34,35,36,37,52,53,54,55,56,57,58,59,60,61,62,63,64,65,66,67,68,69,70,71,72,73,74,75]. CI, confidence interval.

**Figure 4 life-12-00738-f004:**
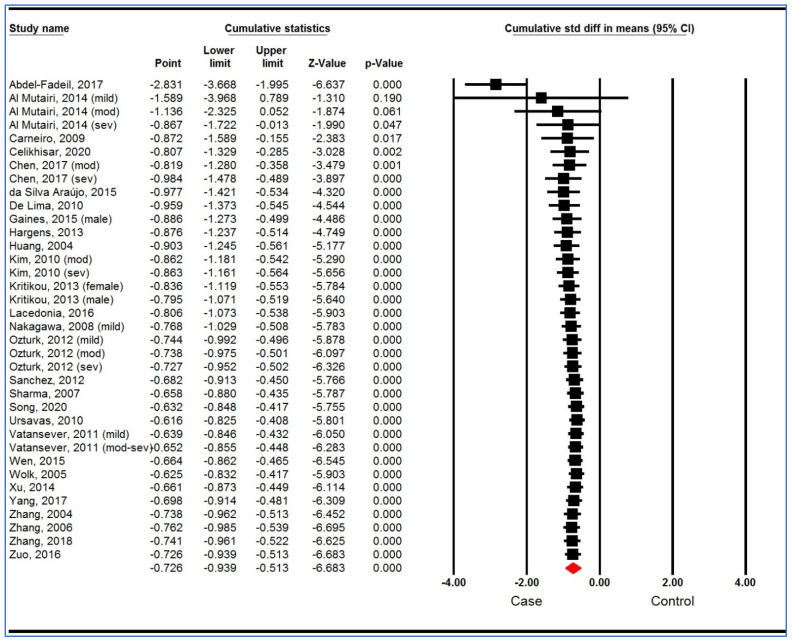
Sensitivity analysis (cumulative analysis) of studies on plasma/serum adiponectin levels for obstructive sleep apnea syndrome cases versus controls [34,35,36,37,52,53,54,55,56,57,58,59,60,61,62,63,64,65,66,67,68,69,70,71,72,73,74,75]. CI, confidence interval.

**Figure 5 life-12-00738-f005:**
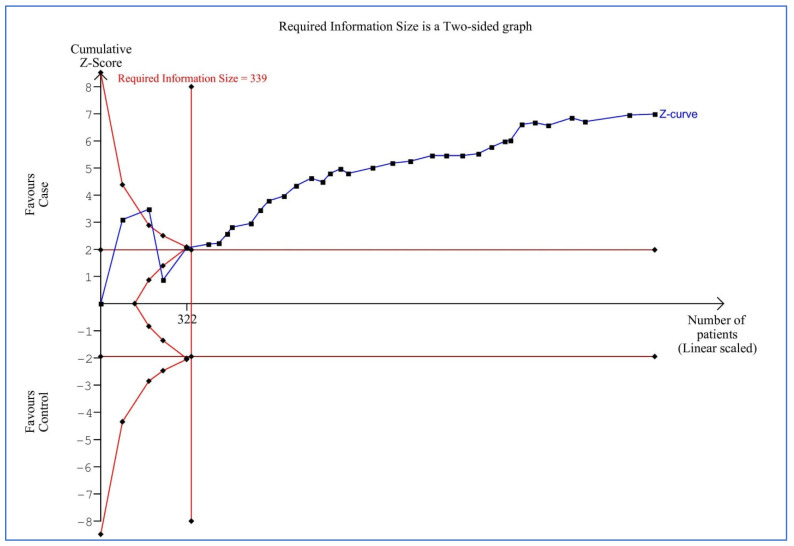
Trial sequential analysis of studies on plasma/serum adiponectin levels for obstructive sleep apnea syndrome cases versus controls.

**Figure 6 life-12-00738-f006:**
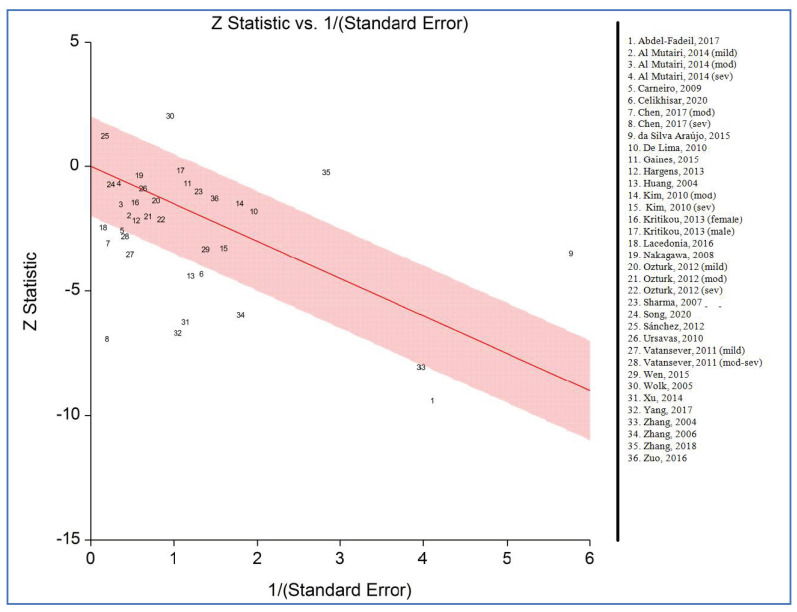
Galbraith (Radial) plot of studies on plasma/serum adiponectin levels for obstructive sleep apnea syndrome cases versus controls [34,35,36,37,52,53,54,55,56,57,58,59,60,61,62,63,64,65,66,67,68,69,70,71,72,73,74,75].

**Figure 7 life-12-00738-f007:**
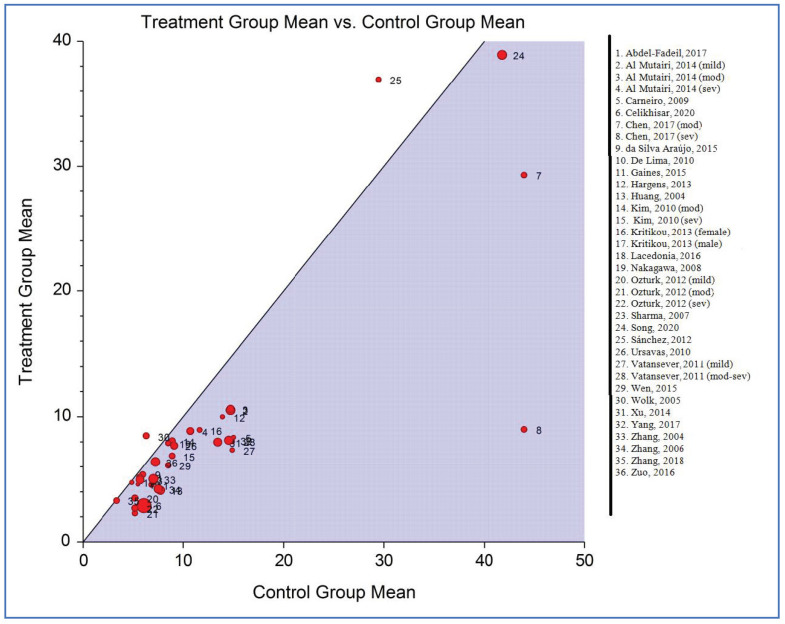
L’Abbé plot of studies on plasma/serum adiponectin levels for obstructive sleep apnea syndrome cases versus controls [34,35,36,37,52,53,54,55,56,57,58,59,60,61,62,63,64,65,66,67,68,69,70,71,72,73,74,75]. Each circle represents each individual study, and the circles are proportional to the study weights (participant number). The diagonal line indicates that the adiponectin mean was equal in the two groups within the studies. Note: For better displaying of numbers in left side, number 18 has been removed.

**Figure 8 life-12-00738-f008:**
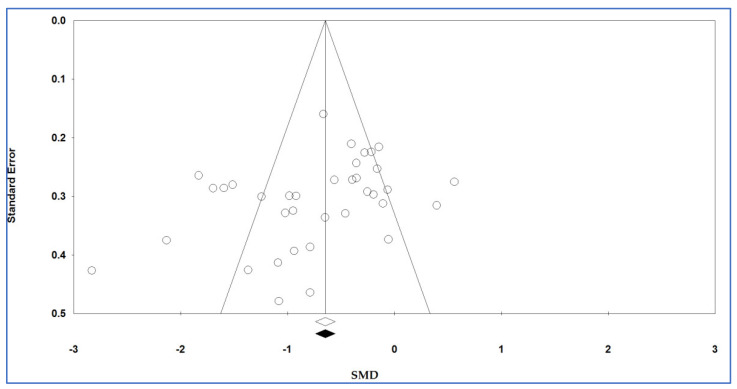
Funnel plot of studies on plasma/serum adiponectin levels for obstructive sleep apnea syndrome cases versus controls. Plot shows the standard error of the standardized mean difference (SMD, Y axis) versus the reported SMD (X axis) using a random effects model. The vertical line indicates the pooled effect estimate. Each circle represents a study. Open diamond represents the pooled effects from the original studies. Black diamond represents the pooled effects incorporating the imputed studies.

**Table 1 life-12-00738-t001:** Characteristics of included articles.

First Author, Publication Year	Country	Ethnicity	Adiponectin Source	Assay Approach	NOS
Huang, 2004 [52]	China	Asian	Serum	RIA	6
Zhang, 2004 [53]	China	Asian	Serum	RIA	6
Wolk, 2005 [36]	USA	Mixed	Serum	RIA	7
Zhang, 2006 [54]	China	Asian	Plasma	RIA	6
Sharma, 2007 [71]	India	Asian	Plasma	ELISA	7
Nakagawa, 2008 [72]	Japan	Asian	Serum	ELISA	7
Carneiro, 2009 [68]	Brazil	Mixed	Plasma	RIA	7
Ursavas, 2010 [64]	Turkey	Caucasian	Serum	RIA	8
De Lima, 2010 [69]	Brazil	Mixed	Plasma	ELISA	7
Kim, 2010 [73]	Korea	Asian	Plasma	RIA	7
Vatansever, 2011 [65]	Turkey	Caucasian	Serum	RIA	7
Öztürk, 2012 [66]	Turkey	Caucasian	Serum	ELISA	6
Sánchez, 2012 [74]	Spain	Caucasian	Plasma	RIA	7
Kritikou, 2013 [62]	USA	Mixed	Serum	ELISA	7
Hargens, 2013 [61]	USA	Mixed	Serum	ELISA	8
Al Mutairi, 2014 [35]	Kuwait	Caucasian	Plasma	ELISA	7
Xu, 2014 [55]	China	Asian	Serum	ELISA	7
Gaines, 2015 [63]	USA	Mixed	Plasma	ELISA	7
da Silva Araújo, 2015 [70]	Brazil	Mixed	Plasma	ELISA	7
Wen, 2015 [56]	China	Asian	Serum	ELISA	7
Lacedonia, 2016 [75]	Italy	Caucasian	Serum	ELISA	7
Zuo, 2016 [57]	China	Asian	Serum	ELISA	6
Chen, 2017 [58]	China	Asian	Serum	ELISA	8
Abdel-Fadeil, 2017 [34]	Egypt	Caucasian	Serum	ELISA	7
Yang, 2017 [59]	China	Asian	Serum	ELISA	7
Zhang, 2018 [60]	China	Asian	Plasma	ELISA	8
Song, 2020 [37]	China	Asian	Serum	ELISA	8
Celikhisar, 2020 [67]	Turkey	Caucasian	Serum	ELISA	8

NOS, Newcastle-Ottawa scale; ELISA, Enzyme linked immunosorbent assay; RIA, Radioimmunoassay.

**Table 2 life-12-00738-t002:** Characteristics of the subjects in the included studies.

First Author, Publication Year	Number: (Case/Control)		Adiponectin (Mean ± SD)		Age (Mean ± SD), Year		BMI (Mean ± SD), kg/m^2^		AHI (Mean ± SD), Events/h	
	Case	Control	Case	Control	Case	Control	Case	Control	Case	Control
Huang, 2004 [52]	69	15	4.12 ± 2.49	7.74 ± 4.42	51.71 ± 12.37	56.37 ± 11.49	26.35 ± 2.60	26.02 ± 1.78	39.28 ± 22.80	3.67 ± 1.38
Zhang, 2004 [53]	71	26	5.03 ± 1.01	7.04 ± 1.31	51.4 ± 11.8	49.2 ± 10.1	26.7 ± 2.1	27.8 ± 1.9	-	-
Wolk, 2005 [36]	26	29	8.49 ± 4.69	6.32 ± 2.96	46 ± 5.10	46 ± 10.77	31 ± 5.10	31 ± 5.39	44 ± 20.40	3.0 ± 2.15
Zhang, 2006 [54]	20	66	4.23 ± 2.04	7.52 ± 2.21	50.7 ± 12.9	49.6 ± 9.2	26.8 ± 2.5	25.6 ± 1.8	36.8 ± 21.4	2.4 ± 1.8
Sharma, 2007 [71]	40	40	4.96 ± 3.21	5.71 ± 3.67	42.3 ± 8.3	43.3 ± 7.8	29.8 ± 3.3	29.1 ± 2.3	32.19 ± 9.94	1.35 ± 0.61
Nakagawa, 2008 [72]	24	18	7.9 ± 5.6	8.5 ± 5.5	45.9 ± 12.6	43.8 ± 12.2	26.7 ± 5.8	23.8 ± 4.0	8.9 ± 2.7	2.4 ± 1.5
Carneiro, 2009 [68]	16	13	8.7 ± 4.72	15.0 ± 9.19	40.1 ± 2.8	38.8 ± 3.3	46.9 ± 2.0	42.8 ± 1.3	65.7 ± 9.9	3.2 ± 0.5
Ursavas, 2010 [64]	55	15	7.7 ± 5.2	9.1 ± 6.6	51.1 ± 8.9	48.4 ± 36.65	32.5 ± 64.11	31.6 ± 7.0	43.5 ± 26.7	2.8 ± 1.5
De Lima, 2010 [69]	10	10	4.6 ± 1.58	5. 5± 0.32	57 ± 33.18	56.8 ± 4.7	34.1 ± 1.3	33.1 ± 7.9	29.5 ± 11.69	3.6 ± 0.32
Kim, 2010 (sev) [73]	28	22	6.88 ± 1.78	8.90 ± 2.63	42 ± 10.77	26 ± 6.91	28.69 ± 4.05	23.88 ± 2.30	52.71 ± 22.23	1.25 ± 1.25
Kim, 2010 (mod) [73]	37	22	8.08 ± 1.66	8.90 ± 2.63	38 ± 15.04	26 ± 6.91	24.43 ± 2.45	23.88 ± 2.30	14.40 ± 4.07	1.25 ± 1.25
Vatansever, 2011 (mod-sev) [65]	9	24	8.04 ± 6.55	14.87 ± 6.17	50 ± 27.11	47 ± 39.19	29.3 ± 4.12	28.4 ± 2.45	54.4 ± 66.79	2.05 ± 1.32
Vatansever, 2011 (mild) [65]	9	24	7.35 ± 2.7	14.87 ± 6.17	48 ± 27	47 ± 39.19	27.6 ± 1.8	28.4 ± 2.45	8.0 ± 2.52	2.05 ± 1.32
Sánchez, 2012 [74]	21	20	36.94 ± 21.42	29.47 ± 15.88	49.33 ± 10.71	42.9 ± 9.16	25.02 ± 1.22	24.71 ± 2.39	41.45 ± 18.3	2.87 ± 1.51
Öztürk, 2012 (sev) [66]	25	32	2.7 ± 3.2	5.2 ± 5.2	50.0 ± 11.7	48.3 ± 10.8	34.0 ± 5.5	31.3 ± 5.6	61.6 ± 20.4	1.8 ± 1.4
Öztürk, 2012 (mod) [66]	12	32	2.3 ± 1.2	5.2 ± 5.2	58.7 ± 8.6	48.3 ± 10.8	32.7 ± 5.8	31.3 ± 5.6	23.4 ± 4.6	1.8 ± 1.4
Öztürk, 2012 (mild) [66]	25	32	3.5 ± 4.2	5.2 ± 5.2	48.8 ± 10.6	48.3 ± 10.8	32.1 ± 6.6	31.3 ± 5.6	9.0 ± 2.6	1.8 ± 1.4
Kritikou, 2013 (female) [62]	20	18	8.96 ± 5.48	11.63 ± 6.17	57.28 ± 6.00	54.21 ± 6.61	31.52 ± 1.54	30.36 ± 2.75	33.94 ± 18.78	1.69 ± 1.61
Kritikou, 2013 (male) [62]	16	13	4.75 ± 2.41	4.89 ± 2.55	53.87 ± 6.76	52.39 ± 6.23	27.09 ± 2.60	26.60 ± 2.65	42.42 ± 22.51	3.03 ± 1.98
Hargens, 2013 [61]	12	18	10.0 ± 2.77	13.9 ± 5.94	22.8 ± 2.77	22.5 ± 2.97	32.4 ± 3.46	31.6 ± 4.67	25.4 ± 18.71	2.2 ± 1.27
Al Mutairi, 2014 (sev) [35]	25	42	8.87 ± 7.41	14.70 ± 13.01	50.0 ± 14.8	55.8 ± 16.3	44.4 ± 16.8	40.5 ± 11.9	48.9 ± 13.1	2.6 ± 1.6
Al Mutairi, 2014 (mod) [35]	29	42	10.64 ± 8.36	14.70 ± 13.01	53.0 ± 16.3	55.8 ± 16.3	49.8 ± 13.9	40.5 ± 11.9	18.0 ± 6.9	2.6 ± 1.6
Al Mutairi, 2014 (mild) [35]	51	42	10.53 ± 7.41	14.70 ± 13.01	49.1 ± 17.0	55.8 ± 16.3	43.3 ± 10.5	40.5 ± 11.9	10.3 ± 3.8	2.6 ± 1.6
Xu, 2014 [55]	62	20	7.98 ± 3.74	13.43 ± 2.04	50.8 ± 11.7	51.3 ± 10.5	-	-	18.03 ± 5.10	2.14 ± 0.89
Gaines, 2015 [63]	47	15	5.13 ± 2.88	5.70 ± 2.94	53.46 ± 0.86	52.70 ± 1.52	27.79 ± 0.35	26.50 ± 0.63	22.94 ±2.47	2.26 ± 4.37
da Silva Araújo, 2015 [70]	33	20	5.38 ± 0.44	5.98 ± 0.82	39.60 ± 1.48	32.50 ± 2.09	34.39 ± 0.51	34.51 ± 0.66	20.16 ± 3.57	2.55 ± 0.35
Wen, 2015 [56]	33	15	6.13 ± 1.16	8.51 ± 3.84	48.5 ± 10.06	46 ± 10.4	27.4 ± 5.0	27 ± 4.6	41.3 ± 3.6	2.8 ± 0.4
Lacedonia, 2016 [75]	10	10	63.43 ± 11.53	80.10 ± 18.50	62.80 ± 8.19	58.80 ± 17.55	27.29 ± 2.41	27.58 ± 1.40	-	-
Zuo, 2016 [57]	59	30	6.4 ± 3.25	7.24 ± 2.42	47.3 ± 14.6	43.3 ± 7.8	27.3 ± 5.0	26.1 ± 1.7	20.6 ± 26.3	-
Chen, 2017 (sev) [58]	35	15	9 ± 13.35	43.98 ± 22.13	42.06 ± 11.75	41.33 ± 12.45	26.12 ± 3.49	25.51 ± 2.17	72.47 ± 15.73	4.86 ± 2.31
Chen, 2017 (mod) [58]	34	15	29.31 ± 11.57	43.98 ± 22.13	43.32 ± 11.95	41.33 ± 12.45	25.35 ± 2.07	25.51 ± 2.17	32.12 ± 10.21	4.86 ± 2.31
Abdel-Fadeil, 2017 [34]	22	22	4.57 ± 0.98	6.85 ± 0.58	49.92 ± 2.10	47.55 ± 2.35	36.00 ± 1.10	36.62 ± 1.14	32.17 ± 20.59	3.72 ± 1.69
Yang, 2017 [59]	20	66	8.13 ± 2.28	14.52 ± 4.10	50.12 ± 11.25	49.30 ± 10.70	24.61 ± 3.80	23.37 ± 3.55	37.87 ± 14.90	-
Zhang, 2018 [60]	30	20	3.30 ± 1.10	3.38 ± 1.39	40.73 ± 8.90	36.10 ± 13.67	28.85 ± 2.62	27.55 ± 2.97	61.48 ± 15.00	1.93 ± 1.38
Song, 2020 [37]	62	33	38.92 ± 18.51	41.80 ± 21.63	40.38 ± 9.50	37.51 ± 10.87	24.18 ± 1.55	23.30 ± 3.07	29.11 ± 20.45	1.78 ± 1.41
Celikhisar, 2020 [67]	84	82	2.9 ± 3.6	6.1 ± 5.8	50.9 ± 5.7	49.3 ± 5.8	32.4 ± 6.0	30.6 ± 5.6	27.4 ± 18.6	1.8 ± 1.4

SD, Standard deviation; BMI, body mass index; AHI, apnea-hypopnea index.

**Table 3 life-12-00738-t003:** Subgroup analysis of plasma/serum adiponectin levels in obstructive sleep apnea syndrome cases versus controls.

Subgroup	Variable (*N*)	SDM	95% CI		Z	*p*-Value	I^2^, %	P_h_
			Min	Max				
Ethnicity	Asian (15)	−0.91	−1.27	−0.56	5.07	<0.00001	84	<0.00001
	Caucasian (13)	−0.63	−0.94	−0.32	4.00	<0.0001	74	<0.00001
	Mixed (8)	−0.41	−0.82	−0.01	2.01	0.04	65	0.006
Country (region)	China (11)	−1.10	−1.54	−0.67	4.98	<0.00001	85	<0.00001
	Turkey (7)	−0.61	−0.81	−0.42	6.09	<0.00001	10	0.35
	USA (5)	−0.15	−0.60	0.30	0.65	0.52	59	0.04
	Brazil (3)	−0.91	−1.33	−0.49	4.27	<0.0001	0	0.93
Adiponectin Source	Serum (23)	−0.85	−1.15	−0.55	5.58	<0.00001	82	<0.00001
	Plasma (13)	−0.47	−0.73	−0.22	3.61	0.0003	64	0.0009
Assay approach	ELISA (25)	−0.69	−0.92	−0.45	5.81	<0.00001	75	<0.00001
	RIA (11)	−0.76	−1.25	−0.28	3.08	0.002	58	<0.00001
Sample size	≥50 (23)	−0.69	−0.94	−0.43	5.24	<0.00001	82	<0.00001
	<50 (13)	−0.77	−1.17	−0.37	3.75	0.0002	74	<0.00001
Publication year	≥2015 (14)	−0.89	−1.25	−0.53	4.83	<0.00001	82	<0.00001
	<2015 (22)	−0.60	−0.87	−0.34	4.43	<0.00001	78	<0.00001
Mean AHI in cases, event/h	≥30 (19)	−0.76	−1.11	−0.41	4.24	<0.0001	84	<0.00001
	15–30 (10)	−0.61	−0.87	−0.34	4.43	<0.00001	61	0.006
	5–15 (5)	−0.42	−0.66	−0.18	3.40	0.0007	30	0.22
Mean BMI, kg/m^2^	BMI ≥ 30 (15)	−0.52	−0.80	−0.24	3.64	0.0003	75	<0.00001
	BMI < 30 (21)	−0.87	−1.16	−0.57	5.69	<0.00001	81	<0.00001
Mean age, y	Age ≥ 50 (18)	−0.79	−1.06	−0.53	5.86	<0.00001	75	<0.00001
	Age < 50 (18)	−0.64	−0.96	−0.31	3.87	0.0001	81	<0.00001

*N*, number of studies; SD, standard deviation; BMI, body mass index; AHI, apnea-hypopnea index, P_h_, P_heterogeneity_; ELISA, enzyme linked immunosorbent assay; RIA, radioimmunoassay.

**Table 4 life-12-00738-t004:** Random-effects meta-regression of plasma/serum levels of adiponectin for obstructive sleep apnea syndrome cases versus controls.

Variable (N)	Point Estimate	Standard Error	Lower Limit	Upper Limit	Z-Value	*p*-Value
Publication year	−0.00627	0.02642	−0.05805	0.04552	−0.23716	0.81254
Sample size	−0.00006	0.00388	−0.00825	0.00694	−0.169166	0.86567
Mean AHI for cases, event/h	−0.00431	0.00651	−0.01708	0.00845	−0.66203	0.50795
Mean BMI for cases, kg/m^2^	0.01046	0.017066	−0.02297	0.04389	0.61354	0.53952
Mean BMI for control, kg/m^2^	0.00337	0.02080	−0.03740	0.04414	0.16182	0.87145
Mean age for cases, y	−0.00344	0.01438	−0.03163	0.02474	−0.23942	0.81078
Mean age for controls, y	−0.00469	0.01294	−0.3005	0.02067	−0.36255	0.71694

N, number of studies; BMI, body mass index; AHI, apnea-hypopnea index.

**Table 5 life-12-00738-t005:** The results of applying the trim-and-fill method.

Value	Studies Trimmed	Fixed-Effects			Random-Effects			Q Value
		Point Estimate	Lower Limit	Upper Limit	Point Estimate	Lower Limit	Upper Limit	
Observed	-	−0.64667	−0.74011	−0.55323	−0.72636	−0.93939	−0.51334	174.28741
Adjusted	0	−0.64667	−0.74011	−0.55323	−0.72636	−0.93939	−0.51334	174.28741

## Data Availability

No new data were created or analyzed in this study. Data sharing is not applicable to this article.

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
