# Peer review of "Evaluation of Plasma/Serum Adiponectin (an Anti-Inflammatory Factor) Levels in Adult Patients with Obstructive Sleep Apnea Syndrome: A Systematic Review and Meta-Analysis"

_life, 2022, doi:10.3390/life12050738_

Round 1
Reviewer 1 Report
This is an interesting meta-analysis to evaluate the correlation between adiponectin levels and obstructive sleep apnoea syndrome. This manuscript has been carefully prepared; however, I have made the following recommendation for the authors to consider to improve this manuscript.
1) A similar meta-analysis was performed in 2018, and the authors claim they intend to provide an update. It is recommended that the authors highlight why such an update is crucial to be published. The 2018 study included 20 papers, and the current study included 28 papers on the same topic. Are there any significant overlap of the selected papers? Is it possible to perform meta-analysis on more recent publications and then compared with previous meta-analysis? It could be a better approach. Or alternatively only include studies published between 2012-2022. I noted that of the 28 studies, most studies were published prior to 2018, and therefore they may have been included in previous meta-analysis. Therefore, earlier studies will effectively have a greater weighing to the meta-analysis. This should be carefully considered.
2) The authors could consider include the critical appraisal for the 28 papers in the supplementary information. How was the quality of the studies been assessed and has studies of low quality be excluded from meta-analysis?
3) Table 2 column 6-7, should it be Age (Mean ± SD), year?
Author Response
Dear Reviewer,
Thank you very much for all your kind efforts. Your comments were very useful to improve the quality of the manuscript. Please find attached the detailed point-by-point-response.
Thank you again for all your kind efforts.

Reviewer 2 Report
Interesting paper. Minor concernings:
- An interesting paper reported there were no statistically significant differences between the OSAS and control groups concerning total cholesterol, triglyceride, low-density lipoprotein (LDL), high-density lipoprotein (HDL), and glucose levels. Adiponectin was lower in the OSAS group at a statistically significant level in comparison with the control group and was related at a statistically significant level to OSAS intensity. Adropin concentration was determined to be higher in the OSAS group at a statistically significant level in comparison with the control group. please cite doi:10.1155/2020/2571283.
- Lipidi peroxidation and inflammationndisorders in Osa patient have been correlated with several cardiovascular and neurodegenerative disoderds. Please discuss and cite doi:10.3390/bs11120180
- The trim and full méthod should be Applied for bias analysis.
Author Response

(The authors gave the same response as above.)
